# Progress in Understanding Metabolic Syndrome and Knowledge of Its Complex Pathophysiology

**Birendra Kumar Jha [1]**, **Mingma Lhamu Sherpa [2]**, **Mohammad Imran [3]**, **Yousuf Mohammed [3]**, **Laxmi Akhileshwar Jha [4]**, **Keshav Raj Paudel [5,*]** and **Saurav Kumar Jha [6,*]**

1   Department of Biochemistry, Janaki Medical College, Tribhuvan University, Janakpur 45600, Nepal; akshatjmc@yahoo.com
2   Department of Biochemistry, Sikkim Manipal Institute of Medical Sciences, Sikkim Manipal University, Gangtok 737102, Sikkim, India
3   Therapeutics Research Group, Frazer Institute, Faculty of Medicine, The University of Queensland, Brisbane, QLD 4102, Australia; mohammad.imran@uq.edu.au (M.I.); y.mohammed@uq.edu.au (Y.M.)
4   H. K. College of Pharmacy, Mumbai University, Pratiksha Nagar, West Mumbai, Mumbai 400102, Maharashtra, India
5   School of Applied & Life Sciences, Uttaranchal University, Dehradun 248007, Uttarakhand, India
6   Research and Development Department, Curex Pharmaceuticals Private Limited, Kavre, Banepa-10, Janagal 45210, Nepal
*   Correspondence: keshavrajpaudel@gmail.com (K.R.P.); saurav.balhi@gmail.com (S.K.J.)

**Abstract:** The metabolic syndrome (MetS), first introduced by Haller in 1975, was sometimes also known as insulin resistance syndrome, syndrome X, and plurimetabolic syndrome. In 1989, it was rechristened by Kaplan as the "Deadly Quartet" based on a consolidation of central obesity, impaired glucose tolerance, dyslipidemia, and systemic hypertension. MetS is positively associated with a pro-inflammatory and pro-thrombotic state, attributed to increased pro-thrombotic and inflammatory marker activity. Moreover, MetS is frequently associated with increased atherosclerotic cardiovascular disease, impaired glucose tolerance, hyperuricemia, obstructive sleep apnea, and chronic kidney disease. Despite concerted endeavors worldwide, the complexity of the pathophysiology of metabolic syndrome still needs to be clearly understood. Currently, therapeutic possibilities are confined to individual therapy for hyperglycemia, hypertension, hypertriglyceridemia, hyperuricemia, regular physical exercise, and a restricted diet. In this review, progress regarding the understanding and pathophysiology of MetS; recent emerging technologies, such as metabolomics and proteomics; the relation of MetS with obesity, diabetes, and cardiovascular diseases; and the association of MetS with COVID-19 are discussed.

**Keywords:** metabolic syndrome; glucose tolerance; pathophysiology; hyperglycemia; diet





## 1. Introduction

The rapid increase in metabolic syndrome (MetS) prevalence is emerging as a significant public health concern worldwide. The upward trend of urbanization, high caloric diet uptake, decreased physical activities, and central obesity compounded with a sedentary lifestyle are assigned as the influential underlying factors contributing to the epidemic upsurge of MetS. The significant increase in the prevalence of MetS and its future challenge to the national and world health scenario is very important. Various epidemiological studies have foisted the conclusion that MetS confers a five-fold increased risk of developing type 2 diabetes mellitus (type 2 DM) and a two-fold increase in the risk of developing a cardiovascular disorder over the next five to ten years [1]. Moreover, an individual with MetS is two to four times more susceptible to developing stroke and at a three- to four-fold risk of progressing myocardial infarction (MI) [2]. These events increase the risk of dying two-fold compared with those without MetS [3]. In 2001, the term MetS became

institutionalized with the ICD-9 (International Code for Diseases-9) code 277.7 and was thought of as a first-order threat for the progression of atherothrombotic complications.

Visceral obesity and insulin resistance (IR) are recognized as the major intrinsic risk factors for MetS. Additionally, decreased physical activities, atherogenic dyslipidemia, calorie-rich dietary intake, and hormonal imbalance are risk factors for developing MetS [4]. The complete pathophysiology of MetS is still unclear. However, redundant adipose tissues-induced persistent low-grade inflammation is considered the crucial underlying cause of developing central obesity-related disorders, such as type 2 DM, cardiovascular diseases (CVDs), and IR [5]. The redundant adipose tissue-induced low-grade persistent inflammatory condition was found to be involved in the progression of diseases related to MetS, such as atherosclerosis, atherogenic dyslipidemia, hypertension, pro-thrombotic status, and impaired glucose tolerance [5]. Herein, we will discuss progress regarding the understanding and pathophysiology of MetS; recent emerging technologies, such as metabolomics and proteomics to deeply understand the pathophysiology of the former; the relation of MetS with obesity, diabetes, and CVDs; and the association of MetS with COVID-19.

## 2. Historical Background

Though Haller first introduced the term MetS in his scientific literature in 1975 [6], which was sometimes also known as IR syndrome and "Syndrome X" by Reaven in 1980 [7], the documented historical evidence of MetS begins with the Italian physician, Morgagni, around four decades earlier [8]. Morgagni noticed a significant association between central obesity, increased arterial blood pressure, atherosclerosis, elevated plasma uric acid, and obstructive sleep apnea. Paulescu observed an interconnection between obesity and diabetes and forwarded the statement "most frequently, obese people become glycosuric" in 1920 [9]. In 1927, a Spanish endocrinologist described hypertension and obesity as pre-diabetic conditions [10]. Since its origin, it has drawn the attention of many researchers worldwide. Later in 1947, Vague developed the concept that central obesity was commonly associated with the metabolic alteration observed in CVDs and type 2 DM [9].

The term plurimetabolic syndrome was introduced in the 1960s to describe the clinical condition of the frequent and concurrent presence of central obesity, dyslipidemia, Type 2 DM, and systemic increased arterial blood pressure [9,11]. In 1965, an abstract was presented at the European Association referencing the study in the diabetes annual meeting, which redefined the syndrome characterized by increased arterial blood pressure, hyperglycemia, and central obesity [11]. It is believed that the MetS field accelerated forward significantly after Reaven's banting lecture, which came up with a cluster of risk factors for DM and CVDs. Reaven was the first to introduce the concept of IR associated with MetS [12]. However, he astonishingly passed over the crucial component of visceral obesity, later considered a pivotal abnormality. Furthermore, regarding the history of MetS, in 1989, it was renamed the "Deadly Quartet" based on the combination of central obesity, increased blood glucose, hypertriglyceridemia, and hypertension described by Kaplan [13], and further, in 1992, it was retitled, the IR syndrome [14].

## 3. Definition and Diagnostic Criteria

The term syndrome derives from the Greek word "Sundromos" (Sun-syn + dromos = to run), meaning to run together. MetS is characterized by a complicated modification of the metabolism, which involves modification of lipid metabolism (dyslipidemia and obesity), carbohydrate metabolism (glucose intolerance), along with increased arterial blood pressure (hemodynamic disturbance with a metabolic starting point) [15]. Recently, pro-inflammatory, pro-thrombotic, and hormonal factors have been reported to be involved in MetS [16]. These modifications of metabolism are interconnected to each other and are found to be involved in increasing the risk of coronary heart diseases (CHDs), cardiovascular atherosclerotic diseases, and type 2 DM and causing mortality [17].

Several documented pieces of evidence reveal that many international organizations and expert groups have attempted to coin the criteria to diagnose MetS. The first attempt was made by the World Health Organization (WHO) in 1998 to define the diagnostic criteria for MetS, followed by the Europe Group for the Study of IR (EGIR), The National Cholesterol Education Program Adult Treatment Panel (NCEP: ATP III) (Table 1), the American Association of Clinical Endocrinologists (AACE), and the International Diabetes Federation (IDF). The first criteria developed by the World Health Organization (WHO), in 1998, with the inclusion of the presence of IR, impaired glucose tolerance (IGT), or type 2 DM as absolutely required factors of MetS with at least two of the following factors: waist/hip ratio—male > 0.9 cm, female > 0.85 cm—or body mass index (BMI) > 30 kg/m$^2$; fasting blood sugar (FBS)—≥110 mg/dL—or IR or type 2 DM or triglycerides (TG) ≥ 150 mg/dL; high-density lipoprotein–cholesterol (HDL-C)—male < 40 mg/dL, female < 50 mg/dL; blood pressure (BP)—diastolic ≥ 140 and systolic ≥ 90 mmHg, respectively; and microalbuminuria [18] (Table 1). Within a year, the European Group for studying IR (EGIR) challenged the above-mentioned diagnostic criteria. It modified the WHO definition by excluding microalbuminuria as an essential component of MetS and included hyperinsulinemia instead [19]. The EGIR considered IR as the substantial cause of MetS and gave much more importance to obesity than the WHO; while excluding persons with type 2 DM, the EGIR defines MetS as hyperinsulinemia or IR along with two extra parameters as FBS ≥ 108.11 mg/dL; BP diastolic ≥ 140 and systolic ≥ 90 mmHg; TG ≥ 150 mg/dL; and HDL-C < 39 mg/dL. Shortly after that, in 2001, the National Cholesterol Education Program Adult Treatment Panel (NCEP: ATP III) released its new criteria for MetS, which included waist circumference, blood lipid level, BP, and FBS (Tables 2 and 3). These differed from both the WHO and EGIR definitions. The NCEP did not consider IR as a mandatory component of the diagnostic criteria and stated that any three of the factors (waist circumference ≥ 102 cm in males and ≥88 cm in females; TG: ≥150 mg/dL and/or on drug treatment; HDL-C < 40 mg/dL in males and <50 mg/dL in females; BP diastolic ≥ 130 and systolic ≥ 85 mmHg and/or on drug treatment; and FBS ≥100 mg/dL and/or on drug treatment) would suffice for a diagnosis of MetS [20].

**Table 1.** Diagnostic criteria of metabolic syndrome elucidated over the years by different organizations.

| Clinical Parameters | Criteria | | | | | | |
|---|---|---|---|---|---|---|---|
| | Central Obesity | FBS | ↑ TG | ↓ HDL-C | ↑ BP | Other | Diagnosed as MetS, If |
| WHO (1998) [19] | Waist/hip ratio Male: >0.9 cm Female: >0.85 or BMI > 30 kg/m$^2$ | ≥110 mg/dL or IR or T2DM or Rx | ≥150 mg/dL | Male: <40 mg/dL Female: <50 mg/dL | Diastolic ≥ 140 and systolic ≥ 90 mmHg | Microalbuminuria | Absolutely required IR plus ≥ 2 criteria |
| EGIR (1999) [20] | WC Male: ≥94 cm Female: ≥80 cm | ≥108.11 mg/dL | ≥150 mg/dL | <39 mg/dL | Diastolic ≥ 140 and/or systolic ≥ 90 mmHg or Rx | | Absolutely required IR plus ≥ 2 criteria |
| IDF (2005) [22] | WC defined in terms of ethnicity specific values | ≥100 mg/dL or Rx | ≥150 mg/dL or Rx | Male: <40 mg/dL Female: <50 mg/dL | Diastolic ≥ 130 and/or systolic ≥ 85 mmHg or Rx | | Absolutely required central obesity plus ≥ 2 criteria |
| AHA/NHLBI (2005) [24] | WC Male: ≥102 cm Female: ≥88 cm | ≥100 mg/dL or Rx | ≥150 mg/dL or Rx | Male: <40 mg/dL Female: <50 mg/dL | Diastolic ≥ 130 and/or systolic ≥ 85 mmHg or Rx | | ≥3 criteria |
| AHA/NHLBI and IDF:2009 [25] | WC defined in terms of population- and country-based specific definition | ≥100 mg/dL or Rx | ≥150 mg/dL or Rx | Male: <40 mg/dL Female: <50 mg/dL | Diastolic ≥ 130 and/or systolic ≥ 85 mmHg or Rx | | ≥3 criteria |

**Table 2.** Ethnic-specific values for waist circumference (IDF guideline 2005) [22].

| Country/Ethnic group | Waist Circumference | |
| :---: | :---: | :---: |
| | **Male** | **Female** |
| Europids | ≥94 cm | ≥80 cm |
| South Asians Based on Chinese, Malay, and Asian–Indian population | ≥90 cm | ≥80 |
| Chinese | ≥90 cm | 80 cm |
| Japanese | ≥90 cm | ≥80 cm |
| Ethnic South and Central Americans | Use South Asian recommendation until more specific data are available | |
| Sub-Saharan Africans | Use European data until more specific data are available. | |
| Eastern Mediterranean and Middle East (Arab) population | Use European data until more specific data are available. | |

**Table 3.** Waist circumference thresholds for abdominal obesity by different organizations.

| Population | Organization | Recommended Waist Circumference | |
| :---: | :---: | :---: | :---: |
| | | Male | Female |
| Caucasian | WHO | ≥94 cm (increased risk) ≥102 cm (still higher risk) | ≥80 cm (increased risk) ≥88 cm (still higher risk) |
| United States, Canada, and European | AHA/NHLBI ("ATP III"), Health Canada, and European Cardiovascular Societies | ≥102 cm | ≥88 cm |
| Japanese | Japanese Obesity Society | ≥85 cm | ≥90 cm |
| China | Cooperative Task Force | | ≥80 cm |
| Middle East, Mediterranean, Europid, Asian, and Sub-Saharan African | IDF | ≥94 cm | ≥80 cm |
| Ethnic Central and South American | | ≥90 cm | |

More than one diagnostic criterion for MetS has created confusion for the diagnosis and research of the former. To address the confusion and with the motto to conclude a single definition for MetS, the International Diabetes Federation (IDF) proposed criteria that could be included in diagnostic criteria and epidemiological studies as well as in research on MetS in April 2005 [21]. Although the pathology of MetS, including its other essential causes, is not entirely understood, central obesity and IR are, however, considered the major causative factors [22]. According to the new IDF definition, for a person to be characterized as having MetS, they essentially have central obesity (defined as waist circumference with ethnicity-specific values) plus any two of the following four factors, i.e., a raised TG level, reduced HDL cholesterol, and raised BP and FBS. In the same year, the American Heart Association (AHA) and the National Heart, Lung, and Blood Institute (NHLBI) of the United States introduced other criteria for MetS with the consideration of central obesity adopted by the IDF. They focused on the inflation of metabolic risk factors. According to the AHA/NHLBI criteria, a person possessing any three criteria of the following five will be characterized as having MetS: waist circumference ≥ 102 cm in males and ≥88 cm in females; TG ≥ 150 mg/dL and/or on drug treatment; HDL-C < 40 mg/dL in males and <50 mg/dL in females; BP diastolic ≥ 130 and systolic ≥ 85 mmHg and/or on drug treatment; and FBS: ≥100 mg/dL and/or on drug treatment [23] (Tables 2 and 3). Moreover,

in 2009, a joint scientific statement by the International Diabetes Federation Task Force on Epidemiology and Prevention; the National Heart, Lung, and Blood Institute; the American Heart Association; the World Heart Federation; the International Atherosclerosis Society; and the International Association for the Study of Obesity was published with a common consensus to define MetS. According to a joint statement, for a person to be characterized as having MetS, they must have any three criteria out of the following five: central obesity (population- and country-based specific definition), TG $\geq$ 150 mg/dL and/or on drug treatment; HDL-C < 40 mg/dL in males and <50 mg/dL in females; BP diastolic $\geq$ 130 and systolic $\geq$ 85 mmHg and/or on drug treatment; and FBS $\geq$ 100 mg/dL and/or on drug treatment [24].

## 4. Prevalence of Metabolic Syndrome

More than 3 million hits on the Google search engine and nearly 10,000 registered articles in PubMed, resulting from a search for "Metabolic Syndrome", are evidence of increased concern regarding MetS [1]. Such increased concern regarding MetS is linked to the increasing prevalence of MetS worldwide. The prevalence of MetS ranges from <10 to 80%, depending upon the region, urban/rural environments, composition (ethnicity, sex, age, and race) of the study population and the accepted definition/criteria for diagnosis of MetS. Regardless of any criteria, the prevalence of MetS is high in all Western societies, which is thought to be the result of the epidemic of central obesity [25]. A study assessed by the National Health and Nutrition Examination Survey (NHANES) in 2003–2006 observed that around 34% of the studied population satisfied the criteria for MetS, with significant association with age [26,27]. As reported by the IDF, one-fourth of the world population has met the criteria for MetS [18]. At the same time, one more study carried out in the United States of America observed nearly 910,000 adolescents qualified for MetS diagnosis [28]. In addition, the global prevalence of MetS of 8 to 43% in males and 7–56% in females was reported by the NCEP-ATP III in 2001 [27]. Furthermore, despite the relatively lower prevalence of obesity in the Asian population, as per the WHO criteria, the rising prevalence of MetS is a significant public health problem compared to the Western world [29]. China, Hong Kong, Taiwan, and Thailand had similar prevalence rates, ranging from 10–15%, whereas higher prevalence was noted in Koreans [30], despite approximately similar BMI. A significantly high prevalence rate of MetS has been reported in India at approximately 31.6% (males 22.9% and females 39.9%) compared to other Asian countries [31]. The prevalence of MetS is increasing significantly in developing counties, varying from 9.8% in the males of north urban Indians to 42% in Iranian urban females [32]. A 74% prevalence of MetS with no significant differences between males (77.7%, 95% CI: 71.0–83.5%) and females (72.2%, 95% CI: 65.2–78.3%) was reported in 2021 in Nepal, using the IDF criteria [33]. Another study in India in 2011, applying the NCEP: ATP III, reported that nearly 95% of the examined population had at least one abnormal parameter, with the overall prevalence of BMI at 79.1% [34]. A survey carried out in 2007 among Asians in China, Hong Kong, Taiwan, Japan, the Philippines, and Singapore reported a 10 to 30% prevalence of MetS, and most subjects had a history of type 2 DM and CVDs [35]. Other studies reported a 14.0 ("WHO"), 26.1 ("ATP III"), 29.8 ("IDF"), and 32.5% ("modified ATP III") mean prevalence of MetS in South Asian regions with a significantly lowered level of HDL cholesterol and hypertension documented as the major risk factor [36].

## 5. Pathophysiology

Despite concerted efforts worldwide, the intricacy of the pathophysiology of MetS is still not clearly understood. Its geographical distribution and increasing prevalence trend in developing countries indicate the association of environmental factors and lifestyle factors, such as high caloric diet intake compounded with decreased/lack of physical activities [37]. There are various hypothetical mechanisms regarding the rudimentary pathophysiology of MetS; however, fatty acid flux and IR, neurohormonal activation, low-grade chronic

inflammation, and oxidative stress are widely accepted as the crucial underlying factors involved in the initiation, progression, and transformation of MetS [38,39] (Figure 1).

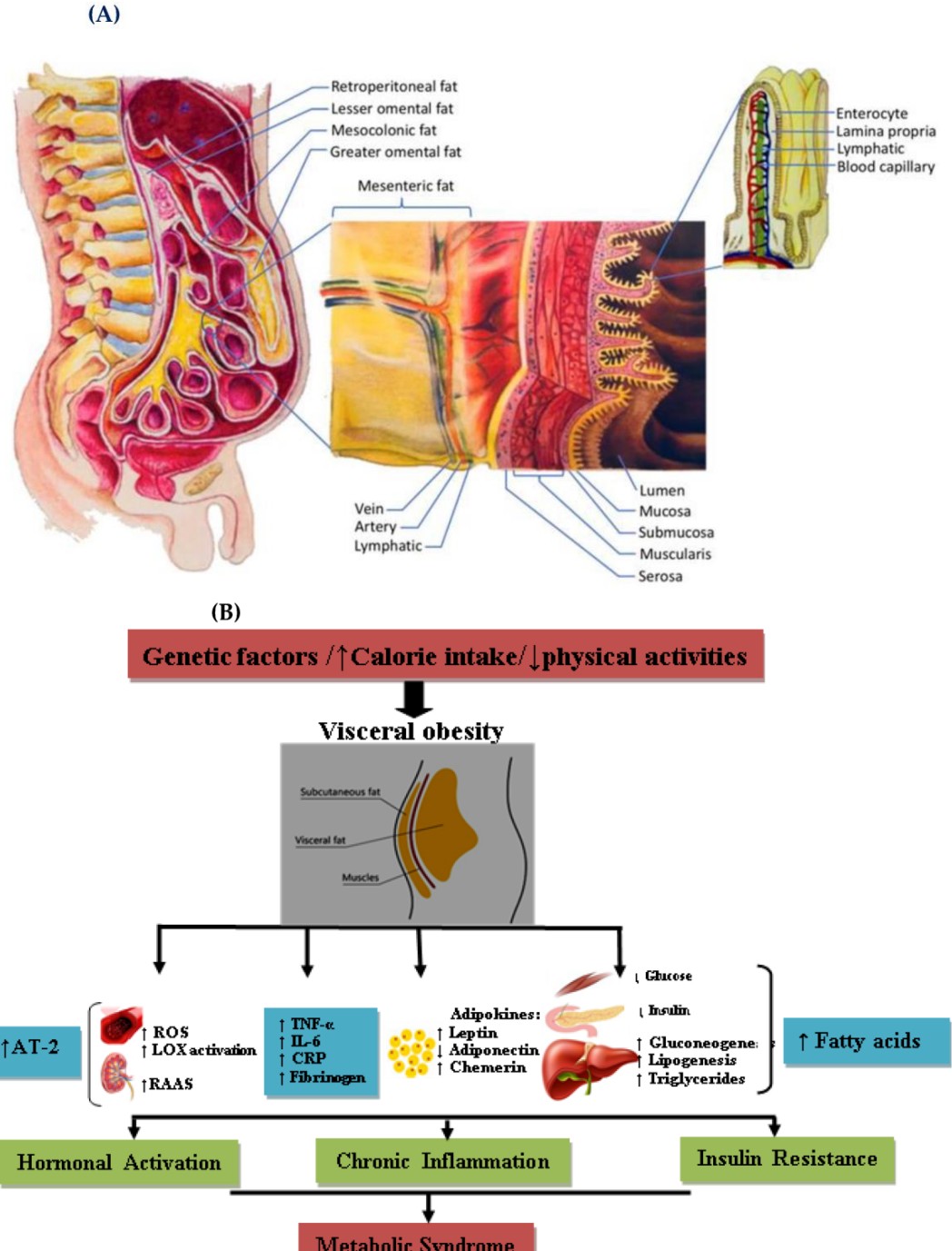

**Figure 1.** (**A**) The accumulation of dietary triglycerides by visceral fat in the abdomen. The enterocytes border the intestinal lumen process and absorb dietary triglycerides. Enterocytes secrete dietary lipids to the lamina propria as VLDLs and chylomicrons. Multiple blood capillaries (red/blue) and lymphatic capillaries within the lamina propria (green). Figure reproduced with permission and without modification from Nauli and Matin [40]. (**B**) Schematic illustration of the pathophysiology of metabolic syndrome. ROS: reactive oxygen species; LOX: lipoxygenase; RAAS: renin–angiotensin–aldosterone system; TNF-α: tumor necrosis factor-α; IL-6: interleukin-6; CRP: C-reactive protein; AT-2: angiotensin-2.

### 5.1. Insulin and IR

Insulin, the first unearthed peptide hormone secreted by β-cells of the pancreatic islet, is characterized as a major anabolic hormone principally involved in the regulation of carbohydrates, fats, and protein metabolism. Insulin acts via its specific receptor, known as the insulin receptor (I-R), which is abundantly present on the cell membrane of the target tissues of the liver, skeletal muscles, and adipocytes. The I-R is a homodimer consisting of two extracellular α-subunits, which facilitate interaction with insulin, and two intracellular β-subunits possess tyrosine kinase enzyme activity. The binding of insulin at the extracellular α-unit of I-R auto-activates the tyrosine kinase enzyme activity on the intracellular domain of the β-subunit. Subsequently, it phosphorylates the protein kinase and I-R substrate (I-RS). Thus, phosphorylated I-RS activates a cascade of signal transduction that leads to the activation of a series of kinases along with transcription factors that mediate the intracellular effects of insulin [40] (Figure 2). The insulin-dependent glucose uptake is mediated by the phosphatidylinositol-3 kinase (PI3) pathway through the Glut4 transporter. A critical downstream effecter protein, protein kinase B, gets activated by protein kinase-3-phosphoinositide-dependent protein 1, in coordination with another currently unidentified kinase named 3-phosphoinositide-dependent protein kinase-2 for now. Thus, the activated Akt eventually phosphorylates and inactivates glycogen synthase kinase-3, promoting glucose storage as glycogen by accelerating glycogenesis [40,41]. In addition to promoting glucose storage, insulin inhibits gluconeogenesis and glycogenolysis and subsequently induces the transcription of enzymes of glycolytic as well as a fatty acid synthetic pathway (Figure 1).

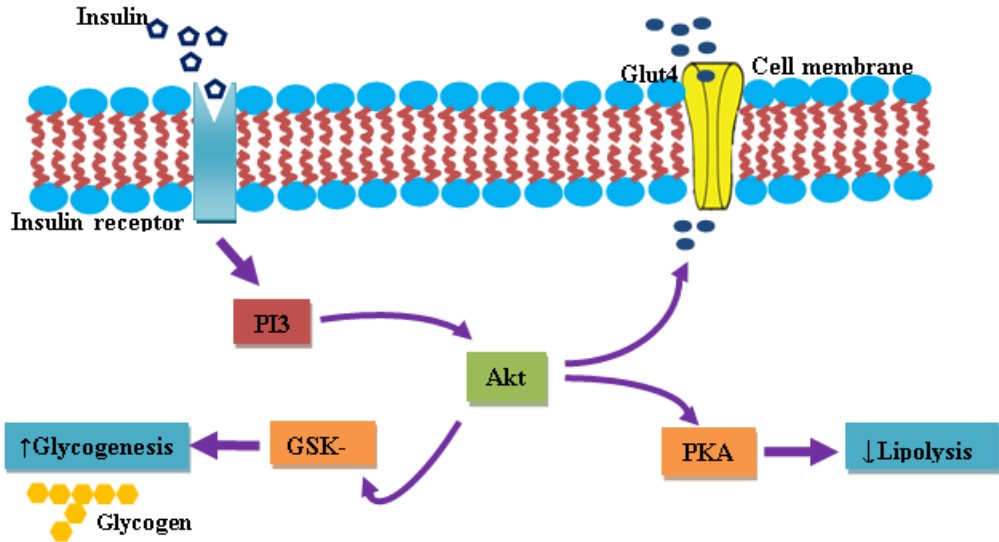

**Figure 2.** Schematic illustration of insulin-dependent glucose uptake and intracellular signaling cascade in the peripheral tissues. PI3k: phosphoinositide 3-kinase; Glut4: glucose transporter 4; PKA: protein kinase A; GSK-3: glycogen synthase kinase 3.

IR-moderated, increased circulating free fatty acids (FFAs) are trusted to play a major role in the pathophysiology of MetS [42]. As maintained earlier, insulin normally stimulates muscle and liver glucose uptake while inhibiting lipolysis and gluconeogenesis. In adipose tissue, IR impairs insulin-associated lipolysis inhibition, resulting in elevated circulating FFAs, further inactivating the anti-lipolytic action of insulin. FFAs decrease glucose uptake in muscle by inhibiting protein kinase activation [43]. In contrast, FFAs increase gluconeogenesis and lipolysis by activating liver protein kinase. Studies have also revealed that FFAs alter the redox status and have a pro-inflammatory action that may be linked to IR and pose difficulty for glucose homeostasis and achieving euglycemic states despite hyperinsulinemia [44].

### 5.2. Adipose Tissue and Its Endocrine Activities

In addition to storing excess energy, adipose tissues function similarly to the endocrine gland and synthesize many biologically active compounds, termed adipocytokines/adipokines, which regulate metabolic homeostasis [26]. A modification in the secretion of pro- and anti-inflammatory adipocytokines in subjects with central obesity might be a major factor contributing to the many aspects of MetS [45]. The secretion of adipocytokines depends on the adipose tissue's mass and energy state. An abnormal secretion of adipocytokines and various inflammatory adipocytokines has been reported in subjects with central obesity. Such abnormal secretion of adipocytokines contributes to a chronic inflammatory condition, ultimately involved in the progression of complications related to MetS and CVDs. The adipocytokines include proteins that interfere with insulin sensitivity (IL-6, tumor necrosis factor-α (TNF-α), resistin, adiponectin, leptin, chemerin, and visfatin), and the proteins affect vascularity (angiotensin and plasminogen inhibitor protein (PAI-1)) [46–48] (Figure 3).

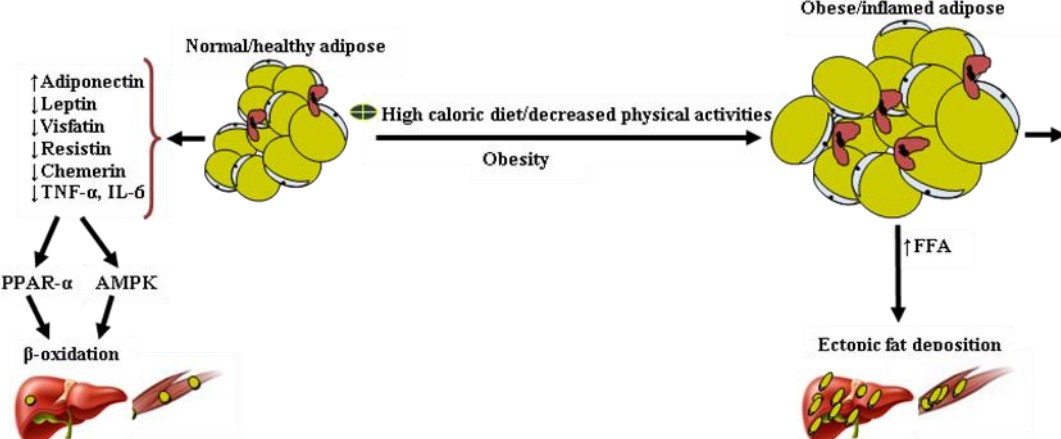

**Figure 3.** Alteration in adipocytokine secretion by obese/inflamed adipose tissue results in ectopic fat deposition. Normal adipose tissue secretes relatively high levels of adiponectin and low levels of pro-inflammatory cytokines. Lipid depositions of the liver and muscles are negligible, maintained by an efficient β-oxidation of fatty acid. In contrast, obese/inflamed and insulin-resistant adipose tissue secretes relatively high pro-inflammatory adipocytokines and low levels of adiponectin. These features and the pro-inflammatory condition may affect the β-oxidation of fatty acids and promote ectopic fat deposition in the liver and muscles.

### 5.3. Leptin

Leptin, the first adipocytokine discovered in 1994 by Friedman and Halaas, is a 167 amino acid protein, including a 21 amino acid signal peptide [49]. The well-known function of leptin is to regulate long-term food intake, body weight, energy expenditure, and neuroendocrine functions. An overexpressed level of leptin is associated with obesity [50]. A lower leptin level has been reported in lean individuals, and a decreasing level has also been observed associated with fasting. Physiologically, leptin promotes the oxidation of FFAs, reducing ectopic fat accumulation in non-adipose tissue, a favored condition for insulin sensitization [51]. A significant alliance has been observed between plasma leptin concentration, CVDs, IR, MetS, hypercholesteremia, and inflammatory markers [52]. Leptin controls energy homeostasis regulated near the hypothalamus called the T helper cell type 1 pathway, the stimulator of immune cell activity [53]. At the molecular level, leptin seals fatty acid entry into mitochondria by malonyl CoA. It inhibits fatty acid oxidation by activating adenosine monophosphate-dependent protein kinase in skeletal muscle [54,55].

### 5.4. Adiponectin

Adiponectin, also named adipoQ, ACRp30, apM1, and GBP28, comprises a collagen tail, a globular head, and an adipocyte-secreted adipocytokine [56]. Adiponectin has been reported as a protective adipocytokine, which exhibits anti-inflammatory and anti-atherogenic properties. The central role of this adipocytokine is to improve IR by accelerating fatty acid oxidation in adipose tissue, resulting in decreased levels of fatty acids in the circulation of the intracellular triglycerides (TG) of the liver and muscle. Furthermore, it has a protective function in the processes of atherosclerosis [57]. At the metabolic level, adiponectin increases insulin sensitivity, mediated through protein kinase A activity, whose activation induces PPARY expression, enzymes of fatty acids oxidation, and other enzymes involved in glucose uptake, in parallel by inhibiting the enzymes of gluconeogenesis [58]. The circulating level of adiponectin has been observed to be adipose tissue mass-dependent and inversely proportional to the mass of adipose tissue [59]. The plasma level of adiponectin was reduced in subjects with MetS, IR, and type 2 DM. Increased plasma adiponectin concentration has been observed to be associated with exercise, weight loss, and thiazolidinedione therapy [60].

### 5.5. Resistin

Resistin is a 114-amino acid circulating protein in the resistin-like family [61]. The precise role of resistin in the human body still needs to unfold. However, it is constrained by growth hormones, glucose, and insulin. An increased resistin expression was reported with impaired glucose uptake and glycogen synthesis [62]. A positive interrelation of resistin with the risk factors of MetS, such as waist circumference, arterial blood pressure, blood glucose, serum triacylglycerol, serum cholesterol, serum VLDL, and plasma insulin, has been observed, which strongly supports the underlying synergy between MetS and resistin secretion as well as central obesity [63]. Furthermore, resistin has been positively related to common inflammatory and fibrinolytic biomarkers, such as CRP, TNF-$\alpha$, and IL-6 [61]. In addition, several recent studies have implicated the association of resistin not only restricted to central obesity but also with CVDs, arthritis, atherosclerosis, and various carcinomas [64].

### 5.6. Visfatin

Visfatin, a pre-B-cell colony-enhancing factor and nicotinamide phosphoribosyltransferase (Nampt), is an adipocytokine of a molecular weight of 52 KDa. Although it is principally secreted by visceral fat tissue, its secretion by other organs, such as bone marrow, activated lymphocytes, liver cells, and skeletal muscle, has also been found. Recently, several studies have observed increased circulating visfatin in individuals with central obesity and type 2 DM [47,65]. Likely, a positive association of visfatin with obesity and IR indicates that visfatin might contribute to the progression of MetS [66]. The visfatin-induced upregulation of pro-inflammatory cytokines, such as Il-6 and TNF-$\alpha$, are thought to play a crucial role in the advancement of IR; however, its complete molecular mechanism is still not fully understood [66].

### 5.7. Chemerin

Chemerin, initially known as TIG2 (tazarotene-induced gene 2), is encoded by the retinoic acid receptor responder 2 (Rarres2) gene, recently recognized as an adipocytokine [66,67]. An increased expression of chemerin has been reported in white adipokines, the liver, and the lungs. In humans, chemerin is initially synthesized as a 63 amino acid precursor, which undergoes further extracellular modification to become active chemerin. Active chemerin exhibits the capability to bind with more than one receptor, such as the G-protein coupled receptor chemokine-like receptor 1(MKLR 1) and G-coupled receptor 1 (GPR 1) and with comparatively low affinity binds to C-C chemokine receptor-like 2 (CCRL 2) [68]. It has been reported that chemerin plays a vital role in adipocyte differentiation, TG synthesis, Glut4 expression, the control of adiponectin, and leptin expression.

Furthermore, an increased chemerin expression in white adipose tissue (WAT), associated with thermogenesis, suggests that chemerin exerts its effect on weight by regulating adipogenesis rather than thermogenesis [69].

*5.8. Inflammatory Status and Oxidative Stress*

The complete mechanism of the onset and progression of MetS has yet to be entirely understood. However, central obesity and IR are reported widely in their etiology, leading to a significant increase in the risk of type 2 DM and CVDs. Chronic low-grade inflammation, oxidative stress, and biologically active adipose tissue are central to its pathophysiology [70]. The oxidative stress associated with obesity and IR leads to the over-activation of the downstream signaling cascade, which is found to be involved in atherogenesis and tissue fibrosis. Secretion of a wide range of inflammatory markers in MetS plays a crucial role in the pathogenesis of CVDs [71]. In addition, increased infiltration of macrophages in inflamed/obese adipose tissue secretes tumor necrosis factor-alpha (TNF-$\alpha$) [72]. TNF-$\alpha$ induced phosphorylation and inactivation of the insulin receptors of the adipose tissue and smooth muscle led to the induction of lipolysis, resulting in increased FFAs with the simultaneous inhibition of adiponectin secretion. IL-6, another vital adipocytokine secreted by adipocytes and immune cells, has shown complex regulatory mechanisms [73]. Increased secretion of IL-6 has also been reported with central obesity and IR, which acts broadly on the liver, bone marrow, and endothelium [74]. The action of IL-6 on these organs results in increased secretion of acute phase reactants by the liver, including CRP. IL-6 increases fibrinogen levels and leads to a pro-thrombotic state; it promotes the secretion of adhesion molecules by endothelial cells and activates the local RAS pathway. Various studies show a strong association between CRP levels and the development of MetS, type 2 DM, and CVDs [75].

## 6. Metabolomics and Proteomics: Emerging Powerful Tools for Precision Medicine to Treat MetS

*6.1. Metabolomics*

A new technique called metabolomics, which is the thorough examination of metabolites in a biological specimen, can guide the practice of precision medicine. Small amounts of metabolites have historically been used to identify both monogenic illnesses, such as inborn metabolic errors and complex metabolic diseases [76]. Metabolomics is the cutting-edge qualitative or quantitative study of metabolites found in biological samples. As metabolic changes are linked with most diseases, either as a cause or because of the disease process, metabolomics is increasingly used in drug and biomarker discovery. Metabolites are necessary components of all living organisms because they provide cellular energy as well as structural and signaling functions. Metabolites, for example, include biochemical classes, such as nucleotides, amino acids, carbohydrates, and lipids, which are the building elements of DNA/RNA, proteins, glycogen, and cellular membranes. Metabolites are both the end products of metabolism (e.g., enzymatic processes) and environmental contaminants (e.g., gut microbiota, diet, and medications). Changes in metabolite levels or composition determine a biological system's phenotype, which is linked with a particular physiological and pathological state. Modern metabolomic technologies can precisely analyze hundreds to thousands of metabolites, far exceeding the capabilities of conventional clinical chemistry methods. As a result, metabolomics allows for the detailed description of metabolic phenotypes. It can facilitate precision medicine on a variety of levels, including the identification of metabolic abnormalities that underlie disease, the discovery of new therapeutic targets, and the identification of biomarkers that may be used to either diagnose a disease or track the activity of treatment. The comprehensive measurement of all metabolites and low-molecular-weight molecules in a biological specimen is the general definition of the emerging discipline of metabolomics. Metabolomics has the potential to be used as a key lens in the molecular microscope for precision medicine because it allows for the profiling of many more metabolites than are currently covered by

standard clinical laboratory techniques, resulting in a complete understanding of biological processes and metabolic pathways. As metabolomics seeks to measure molecules with different physical properties than genomic and proteomic methods, it poses a significant analytical challenge in practice (e.g., ranging in polarity from very water-soluble organic acids to very non-polar lipids [77]. As illustrated in Figure 4, comprehensive metabolomic technology platforms usually employ dividing the metabolome into subsets of metabolites (often based on compound polarity or structural similarity) along with creating distinctive preparation of samples and analytical methods tailored for each subgroup. As a result, the metabolome is measured as a collection of results from different analytical techniques. Metabolomic techniques persist in advancing and changing as an evolving discipline, which has been made possible, at least in part, by the consistent development of analytical instrumentation with novel features every year [78]. Although a drawback of metabolomic laboratories is that they employ numerous procedures that may be frequently improved, each laboratory has distinctive techniques, and there are only a few standard operating procedures (SOPs) that are frequently employed throughout laboratories. However, while the range of technologies is associated with advancement in the field, it can make comparing data from different laboratories challenging due to concerns such as various measurement precision for specific classes of metabolites or non-overlapping metabolite coverage. The level of certainty in metabolite identification can also vary depending on the technique, ranging from putative markers using reference databases to signals still deemed unknown to metabolite identities that have been thoroughly verified using reliable reference standards. Metabolomic practitioners have recognized the need for standardization, which has sparked several initiatives targeted at achieving this objective, such as the Metabolomics Standards Initiative to develop data reporting guidelines [79].

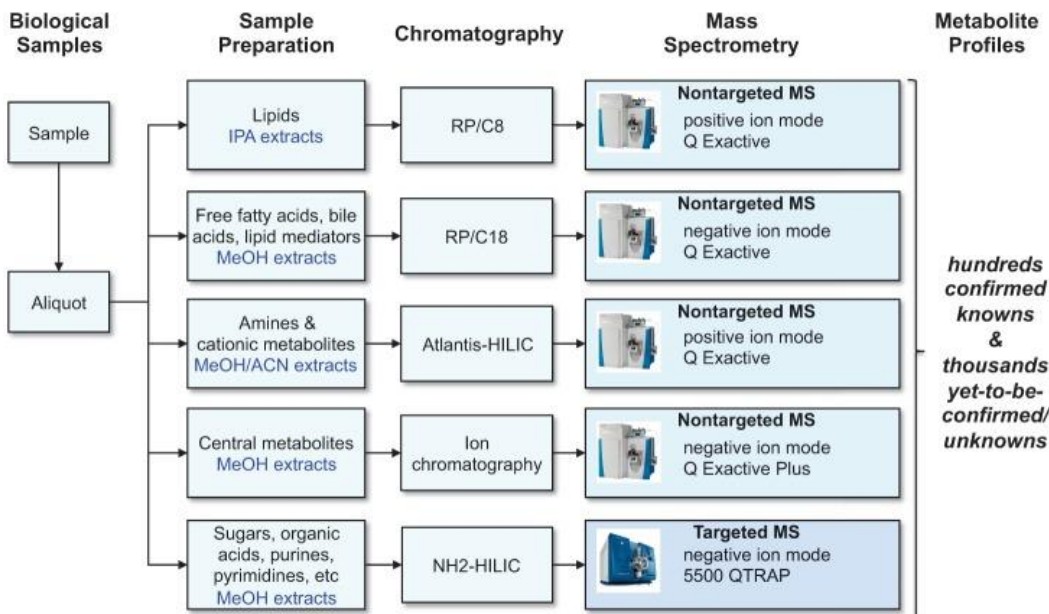

**Figure 4.** An illustration of the metabolomic technology used at the Broad Institute of MIT and Harvard based on liquid chromatography–mass spectrometry (LC-MS). To measure lipids and metabolites of intermediate polarity, such as free fatty acids, bile acids, and polar metabolites, our laboratory has developed a complete metabolomic platform that uses both targeted and non-targeted LC-MS techniques. HILIC: hydrophilic interaction liquid chromatography. IPA: isopropanol. Figure reproduced with permission and without modification from Clish C.B. [77].

*6.2. Proteomics*

The total number of proteins found in a cell, tissue, or organism at any particular time is referred to as the proteome. Due to variations in gene expression, this protein population

can change over time, depending on the development environment and the type of cell. To ascertain the role of protein complexes, proteomics analyzes the interactions, modifications, and activities that take place there. Proteins are fundamental to the functioning of cell systems; they support metabolic processes, provide structural support, and play a crucial role in regulating gene expression by serving as signal receptors or initiators. They make up the end products. For proteome research, mass spectrometry is frequently used. This protein identification technique, which is typically used in conjunction with a liquid chromatography system, depends on fragment detection and measurement, which, when compared to large-scale databases, can precisely identify the peptide sequences present in samples. Protein extraction, enzymatic digestion, HPLC separation, tandem mass spectrometry (LC-MS/MS) analysis of the resulting peptides, database searching, and software-based protein quantification comprise typical proteomic procedures (Figure 5). It is possible to add more stages to either further fractionate the samples or enrich particular components. Proteomics is a particularly promising method to learn about rare genetic diseases, identify biomarkers to help with early diagnosis, and better understand the underlying pathophysiology to influence the development of novel treatments. To provide more precise patient care, the emerging idea of personalized medicine depends on global and integrative approaches (such as proteomic technologies), which can also offer useful information about diseases. Clinical proteomics is the use of proteomic methods to treat illnesses. This represents a potent instrument for investigating changes in disease-related pathways and discovering novel protein biomarkers. A proteomic biomarker is defined as "a specific peptide or protein associated with a particular ailment, such as the onset, manifestation, or progression of a disease or a response to treatment" or "a feature that is measured as an indicator of normal biological processes, pathogenic processes, or a response to an exposure or intervention" [80,81].

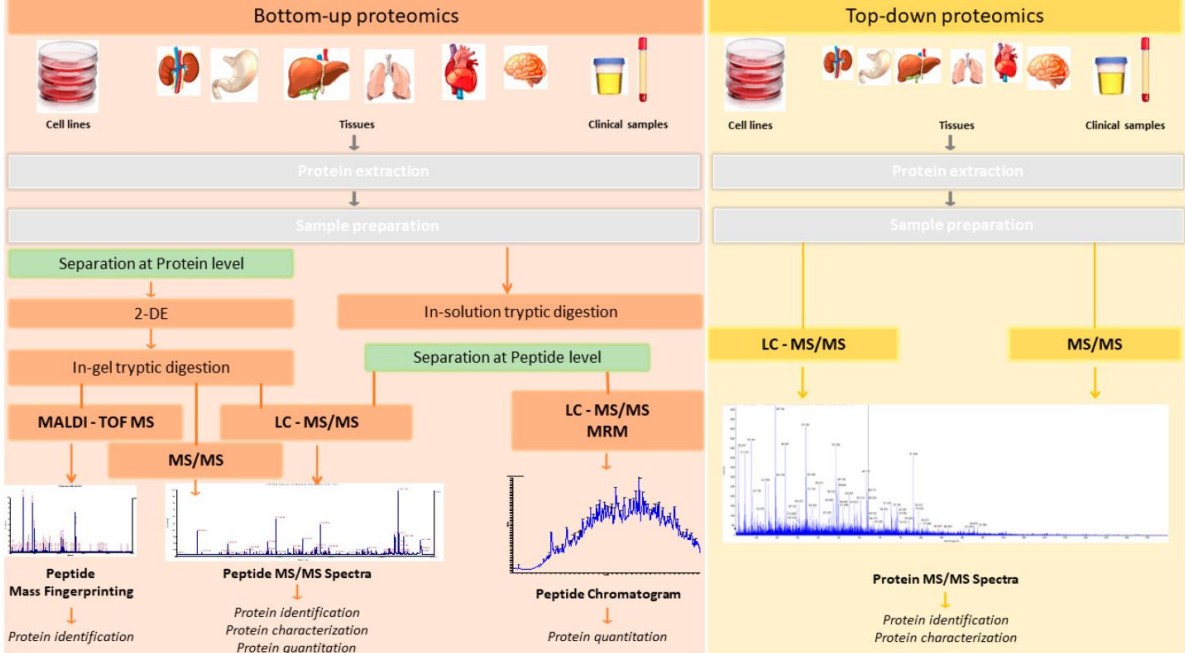

**Figure 5.** Proteomic approaches from the "top-down" and "bottom-up" in clinical data. Proteins are broken down by an exogenous protease (such as trypsin) in "bottom-up" proteomics methods, resulting in internal peptides with suitable length, ionization, and fragmentation parameters for LC MS/MS and database queries. By analyzing these peptides using shotgun proteomics techniques, the source protein can be determined based on the endogenous cleavage site, and the protein can then be identified. Top-down proteomics, in comparison, allows for both methods: direct protein identification without cleavage and protein cleavage with proteases. Figure reproduced with permission and without modification from Chantada-Vázquez et al. [81].

## 7. Metabolic Syndrome and Its Association with CVDs

MetS is characterized as accumulating multiple known cardiovascular risk factors, such as type 2 diabetes, hypertension, high triglycerides, and low levels of high-density HDL-C in the general population [82–86]. The two underlying diseases of the syndrome, central obesity and IR, are additional risk factors for CVDs. MetS also includes several other (non-conventional) risk factors as additional characteristics. They include indicators for thrombophilia (such as plasminogen activator inhibitor 1, PAI1), endothelial dysfunction (such as E-selectin), elevated oxidative stress, and mild to chronic inflammation (such as C-reactive protein or CRP). Consequently, people with MetS may be more prone to atherosclerosis and cardiovascular events. Recent longitudinal studies have found that those with metabolic syndrome develop atherosclerosis at a younger age and have a higher risk of myocardial infarction and stroke than people without the syndrome. Women with the syndrome, as well as people with a history of diabetes, CVDs, or elevated CRP, are particularly vulnerable to being at greater risk of CVDs. On the other hand, people with an accumulation of several minor anomalies are already at a higher risk. When people with the syndrome are compared to people who do not have any metabolic abnormalities, the risk associated with the syndrome is undeniably greater.

### 7.1. Impaired Glucose Regulation and CVDs

Growing evidence suggests that the post-prandial state plays a significant role in the progression of atherosclerosis. While fasting glycemia is within the reference range in patients with impaired glucose tolerance, the post-prandial period is marked by a sharp and significant rise in blood glucose levels. Recently, there has been a lot of focus on the potential link between this post-prandial "hyperglycemic spike" and the onset of CVDs in these people. As the glycemia level after the glucose challenge is a direct and independent risk factor, epidemiological studies have exhibited that impaired oral glucose tolerance is associated with an increased risk of CVDs, even though the oral glucose tolerance test is highly non-physiological. Most cardiovascular risk variables are altered in the post-meal period and are consequently impacted by an abrupt rise in blood sugar levels. The creation of free radicals, which encourages the development of endothelial dysfunction, a prothrombotic condition, and a pro-inflammatory condition, may be the mechanism by which acute hyperglycemia exerts its effects. Future research may assess if the prevention and treatment of CVDs in these people can include reducing post-prandial hyperglycemia in the impaired glucose tolerance stage [87]. Pre-diabetic people are urged to start changing their lifestyles and be informed about the cardiovascular effects of the illness [88]. Most earlier studies that evaluated the relationship between pre-diabetes and CVDs relied on a single blood glucose measurement taken at the time of recruitment rather than monitoring changes in blood glucose levels over time [88]. The remaining query is whether the cause of this link results from the direct effect of pre-diabetes or is mediated by the progression of pre-diabetes to a diabetes state and whether the risk could be reduced by regression from pre-diabetes to normoglycemia. This issue has been investigated in a few studies involving populations from Europe, Korea, and America; some revealed an elevated risk of CVDs in the presence of impaired fasting glucose (IFG) and/or impaired glucose tolerance (IGT) per se (i.e., without changing to a diabetes state) [89,90]. In contrast, others revealed this risk was only elevated after type 2 diabetes had progressed [91]. Previously, following a 7-year follow-up, it was discovered that IFG/IGT was only linked with a 56% increased risk of CVDs in women [92]. Therefore, taking into account the high prevalence and incidence of IFG [93] as well as the high CVD burden in the Middle East and North Africa (MENA) populations [14], it was sought to determine whether a 3-year period of time spent in the IFG, IGT, and IFG/IGT (IFG and/or IGT) states, regression to normoglycemia, or conversion to diabetes state is associated with the long-term risk of CVDs and coronary heart disease (CHD) [94].

### 7.2. Obesity, Central Fat Distribution, and CVDs

Obesity is regarded as a pandemic of the twenty-first century. It is linked to serious chronic non-communicable diseases, particularly CVDs, the leading cause of mortality worldwide. Ectopic fat depots typically localize in visceral adiposity, which raises the risk of CVDs. The increased risk of incident coronary artery disease and atherosclerosis in obesity are both explained by endothelial dysfunction. Additionally, microvascular disease brought on by chronic inflammation raises cytokines and lowers nitric oxide [86]. Chronic inflammation is also characterized by an imbalance between the endothelium's pro-inflammatory and pro-coagulant activities, which results in a pro-coagulant state. The gut microbiota and how it affects the development of atherosclerosis is a crucial subject. According to research, gut dysbiosis can affect the development of atherosclerosis and CVDs. BP, cholesterol, and glucose levels have also been linked in studies to overweight and obesity and coronary artery disease; however, ectopic fat deposition, particularly in the pericardial and epicardial spaces, is an important additional etiological factor that may increase the burden of coronary atherosclerosis. Therefore, to detect, identify, and treat cardiovascular problems in obesity, a great deal of knowledge is required [95]. In contrast to non-obese people, obese individuals have CVDs at a higher rate [96]. Although there has been much discussion regarding the independent contribution of obesity to CVDs and as obesity is not a risk factor in and of itself independent of the problems that frequently accompany excess body fat, the real question is whether obese people have an increased risk. T2DM, dyslipidemia, and hypertension are frequently associated with obesity [97,98], and these classic risk factors usually convey a higher cardiovascular risk. However, some studies suggest that obesity also plays a part independent of these characteristics, which reinforces the idea that carrying too much weight is harmful and should be avoided at all costs [99,100]. The initial research on central fat distribution and CVDs was undertaken more than 20 years ago in Sweden. These investigations revealed that people with high waist-to-hip ratios (WHR) were more likely to develop CVDs [101] and T2DM [102], which some experts consider to be a particular kind of CVD [103]. Individuals with a predominance of central fat have a constellation of metabolic, hemodynamic, pro-coagulant, and IR problems [104]. As a high waist measurement indicates an excess of visceral adipose tissue, the fat depot seems to convey the greater risk of developing metabolic abnormalities and cardiovascular events, and attention has recently been on the waist as a risk factor.

### 7.3. Hypertriglyceridemia and CVDs

Targeting HDL-C has received renewed attention due to residual cardiovascular risk and the failure of current treatments. Triglyceride-rich lipoproteins and residual cholesterol have all been shown to be significant factors to be considered to treat CVDs. Experimental, genetic, and epidemiological investigations have shown that these are risk factors for CVD progression [105,106]. Fibrates, with or without statins, lower the incidence of cardiovascular events. Omega-3 fatty acids at high doses are still being studied, and new specialized triglyceride routes are being targeted for treatment, such as by inhibiting apolipoprotein C-III and angiopoietin-like proteins. LDL is the focus of current prevention recommendations; LDL cholesterol reduction is the primary therapy goal, usually achieved by statins [107–109]. Despite this, many people who take statins still develop cardiovascular problems [110]. Even with low LDL cholesterol, when certain levels (70–100 mg/dL) are met, the remaining cardiovascular risk exists with statins taken at high doses, as seen in randomized clinical trials. Physicians should take additional measures to reduce lingering cardiovascular risk, pay attention to risk variables that can be altered, such as high triglycerides., cholesterol, apolipoprotein B, IR, visceral fat, and low HDL cholesterol. It is significant to note that new genetic studies and randomized trials have shown that hypertriglyceridemia and residual cholesterol may be the cause of CVDs rather than low HDL cholesterol [111]. Following the development of statins, clinical attention was primarily directed toward lowering LDL cholesterol, followed by the potential to raise HDL cholesterol, with less emphasis on lowering triglycerides. However, the realization

from genetic research and unfavorable findings from randomized trials showed that low HDL cholesterol could not be the direct cause of cardiovascular illness, as previously believed, has sparked a renewed interest in high triglyceride levels. Epidemiological and genetic data suggesting elevated triglycerides, residual cholesterol, or triglyceride-rich lipoproteins as an additional cause of CVDs and all-cause mortality have further stoked this increased interest. Triglyceride levels can be evaluated in both non-fasting and fasting conditions. Levels between 2 and 10 mmol/L are associated with an elevated risk of cardiovascular disease, while levels above 10 mmol/L are associated with an increased risk of acute pancreatitis and, perhaps, CVDs. However, there are not many randomized studies confirming the cardiovascular benefits of triglyceride reduction; large-scale trials have already begun, and new triglyceride-lowering medications are being developed. These trials will provide conclusive proof of whether lowering triglycerides lowers the risk of CVDs.

### 7.4. Hypertension and CVDs

Hypertension is the most significant risk factor for all forms of acquired CVDs, including coronary disease, left ventricular hypertrophy, valvular heart disease, cardiac arrhythmias, such as atrial fibrillation, cerebral stroke, and renal failure. Due to the ongoing connection between BP and cardiovascular and renal events, the difference between high normal BP and hypertension is based on arbitrary cut-off numbers for BPs. Overall, 30–45% of the general population with various hypertension is prevalent in European countries, with the prevalence increasing sharply with age. CVD prevention and treatment recommendations should be linked to overall cardiovascular risk, which can be evaluated using various models. Despite having many major risk factors and a significant increase in relative risk, young people (particularly women) are unlikely to achieve high-risk status due to the powerful influence of age on risk. Due to significant national differences, age-adjusted models, models that evaluate relative risks compared to people of a similar age, models that include comprehensive evaluations of target organ damage and ambulatory 24 h BP, and national models are needed [112].

### 7.5. Microalbuminuria and CVDs

Following the early discovery of atherosclerotic markers, therapeutic options for CVDs are increasingly focused on preventive interventions. The section emphasizes microalbuminuria, which is considered a straightforward indicator of an atherogenic environment. Prospective and epidemiologic studies in groups of individuals with diabetes or high BP, as well as in the general population, have suggested that microalbuminuria is predictive parameter of all-cause, and cardiovascular mortality and CVDs progression, independent of established risk factors. The mechanism underlying the connection between albumin excretion and CVDs is unknown. Microalbuminuria is thought to be a sign of CVDs risk because it suggests subclinical vascular injury in the kidneys and other vascular beds. It could also signify widespread endothelial dysfunction, putting them at risk for future cardiovascular issues. According to this theory, routine testing for microalbuminuria may help stratify overall cardiovascular risk and allow early detection of vascular illness, especially in individuals with risk factors like diabetes or hypertension. An intensive multifactorial strategy, including behavior modification and targeted pharmacotherapy, may be required if a urine test for urinary albumin excretion is positive to stop further renal decline and improve the CVD risk factor profile. Angiotensin-converting enzyme inhibitors or angiotensin II receptor blockers, statins, and/or rigorous glycemic management (in diabetics) may significantly reduce cardiovascular and/or renal morbidity in individuals with albuminuria, according to the data from intervention trials. Adopting this (old) marker may enable more effective use of drugs and secondary preventative techniques [113].

## 7.6. Association between IR and CVDs

A vital hormone called insulin controls cellular metabolism in numerous tissues throughout the body. IR is distinguished by abnormalities in glucose uptake and oxidation, a reduction in glycogen synthesis, and, to a lesser degree, the ability to suppress lipid oxidation. IR is described as a decrease in tissue responsiveness to insulin stimulation. On the other hand, the heart's metabolic network is extremely adaptable and can utilize a wide range of substrates, such as glucose, lactate, and amino acids. The literature has widely documented that free fatty acids are the primary substrate for ATP generation in the adult heart. CVDs evolve as a consequence of numerous metabolic changes caused by IR. For example, extended hyperglycemia caused by IR can cause an imbalance in glucose metabolism, which causes oxidative stress and an inflammatory response that damages cells. IR causes the well-known lipid triad—high plasma triglyceride levels, low levels of high-density lipoprotein, and the appearance of small, dense low-density lipoproteins—which can alter systemic lipid metabolism. This triad and endothelial dysfunction contribute to the formation of atherosclerotic plaque, which can also be caused by abnormal insulin signaling. In relation to the systemic effects associated with IR and the metabolic cardiac alterations, it can be concluded that IR in the myocardium causes damage via at least three different mechanisms, including altered signal transduction, impaired regulation of substrate metabolism, and altered delivery of substrates to the myocardium [114].

## 7.7. Fibrinolysis in Acute and Chronic CVDs

Worldwide, the development of an obstructive thrombus within an artery continues to be a leading cause of mortality and morbidity. Modern anti-platelet treatments effectively reduce platelet function but cannot completely remove the risk of atherothrombotic events. This may be due to severe vascular disease, which goes beyond what can be protected by the medications being utilized. Recent research, however, indicates that poor fibrin clot lysis may be partially responsible for the residual vascular risk in patients using contemporary anti-platelet treatments [115]. Therefore, myocardial infarction or stroke, the two main causes of death in Western countries, frequently arise due to pro-coagulant conditions and/or defective fibrinolysis. Among the many possible indicators of a pro-coagulant condition and compromised fibrinolysis are fibrinogen and PAI-1. Numerous studies have noted that elevated fibrinogen levels indicate increased cardiovascular risk [116]. High PAI1 concentrations have also been found to increase the risk of CVDs [117,118]. Additional coagulation factors, such as factors VII and VIII, can forecast future cardiovascular events.

## 7.8. Uric Acid and CVDs

Hyperuricemia is widely underappreciated as a major risk factor for CVDs. However, several studies in the past few decades have shown that uric acid (UA) plays an important part in hypertension, MetS, heart failure, coronary artery disease, diabetes, chronic kidney disease, and overall cardiovascular mortality. Several pathways, including UA-mediated oxidative stress, systemic inflammation, endothelial dysfunction, and renin–angiotensin–aldosterone system activation, have been proposed to explain the involvement of UA in cardiovascular disorders. Most European and foreign guidelines recommend treating symptomatic hyperuricemia with a serum UA target level of lesser or equal to 6 mg/dL. Whether to treat asymptomatic hyperuricemia remains hotly debated. In humans and great apes, UA is a byproduct of purine metabolism. Biological fluids have a 50% overall anti-oxidant capacity in humans, and UA functions as an anti-oxidant. UA transforms into a pro-oxidant agent and causes oxidative stress when it is present in the cytoplasm of cells or the acidic/hydrophobic environment in atherosclerotic plaques and by this mechanism contributes to the pathophysiology of human disease, including CVDS. The majority of epidemiological studies—but not all of them—suggested a connection between elevated serum UA levels and CVDs, including atrial fibrillation, congestive heart failure, arterial hypertension, stroke, and coronary heart disease (CHD), as well as a higher risk of CVD-related mortality in the general population and in individuals with CHD that has

been confirmed. A correlation between increased UA and conventional cardiovascular risk factors, MetS, IR, obesity, non-alcoholic fatty liver disease, and chronic kidney disease has also been shown by the available evidence. The detrimental effects of elevated UA levels on cardiovascular health have been linked to a number of mechanisms, including increased oxidative stress, decreased nitric oxide availability, endothelial dysfunction, the promotion of local and systemic inflammation, vasoconstriction and proliferation of vascular smooth muscle cells, IR, and metabolic dysregulation. However, UA may be pathogenic and contribute to the pathophysiology of CVDs by acting as a bridging mechanism, mediating (enabling), or amplifying the negative effects of cardiovascular risk factors on vascular tissue and myocardium, even though the causality in the relationship between UA and CVDs is still unknown [119,120].

## 8. T2DM and MetS

The prevalence of T2DM, a metabolic condition endemic to the entire world, is rising. Additionally, it is predicted that 366 million individuals will have diabetes by the year 2030. It depends on a sedentary lifestyle, an aging population, etc. Sedentary behavior, bad eating habits, genetics, old age, stress, etc., all raise the risk of T2DM [121]. It happens when the beta cells of the pancreas are damaged, which causes a decrease in insulin production or a lack of insulin receptor sensitivity, which prevents insulin from binding to its receptor effectively. While type 1 diabetes mellitus is an autoimmune disorder in which antibodies target the pancreatic beta cells, it also causes markedly reduced insulin production [122]. Both types of diabetes mellitus are characterized by hyperglycemia, which requires insulin to maintain glycemic levels after regular exercise; a low-calorie diet and oral hypoglycemic medications are insufficient to assure glycemic control [121]. In reaction to elevated blood glucose levels, insulin is generated. This occurs to facilitate glucose's entrance into cells to generate energy [122]. Insulin plays a vital role in the metabolism of lipids, proteins, and carbohydrates. It also facilitates hepatic glycogen production and prevents the liver's gluconeogenesis and lipid breakdown. It controls lipid oxidation as well [96]. Insulin drawbacks include the lack of data on individual glucose levels in diabetics, hypoglycemia, skin necrosis, nerve damage, and pain from subcutaneous injections that lower low compliance [123,124]. Due to its non-invasive nature and higher usage compliance, oral insulin is a superior option to subcutaneous injections. Similarly, despite numerous attempts, oral insulin has yet to be proven to be very effective as it has weak membrane permeability, low bioavailability, and early breakdown [125]. Even a range of drug delivery systems have been designed to deliver the large molecules [126–128]. The present treatments seek to maintain glycemic control, but they have several drawbacks, such as poor target specificity, undesirable side effects, and ineffective doses [129,130]. The aberrant rise of blood glucose produced by a malfunctioning feedback loop between insulin action and secretion is a key component of the pathogenesis of T2DM. It is well known that increasing the amount of glucose produced by the liver while simultaneously reducing the amount of glucose absorbed by tissues, muscles, etc., can raise blood glucose levels; however, cell dysfunction can also prevent the body from producing glucose levels [131,132].

## 9. Obesity and MetS

Obesity is a global health issue and a risk factor for metabolic illnesses such as cancer, type 2 diabetes mellitus, and cardiovascular disorders. Overweight and obesity were blamed for over 4.5 million deaths worldwide in 2013 [133]. Although there are many factors that contribute to obesity, lifestyle and genetics are crucial [134]. Obesity susceptibility genes are thought to exist, and it is thought that the environment interacts with a genetic vulnerability to cause obesity. According to the "thrifty gene" theory, genes that increase energy intake were preferred to increase energy expenditure [135]. The genes FTO and TMEM18 are linked to a higher risk of obesity. The endoplasmic reticulum-associated degradation system mediated by the C2-domain protein AIDA may represent an "an-

tithrifty" mechanism that deactivates the enzymes responsible for absorbing and storing fat. In treating obesity, genetic pathways are also being researched. For example, sterol regulatory element-binding protein (SREBP) has been linked to the activation of genes related to cholesterol synthesis. The uptake of fatty acids and their storage in adipocytes are also stimulated by fatty acid-binding proteins (FABPs) [136]. Additionally, the pathophysiology of obesity is significantly influenced by epigenetics, which involves altering genes through chromatin remodeling, histone modification, non-coding RNA change, and methylation [136]. There are signs that these epigenetic changes may be passed on to succeeding generations, increasing the prevalence of obesity worldwide. The lifestyle-based approach to treating obesity involves cutting back on calorie-dense meals, giving up a sedentary lifestyle, and engaging in regular exercise. While using anti-obesity medications, such as lorcaserin and orlistat, is a pharmacotherapeutic strategy for managing obesity, surgical therapies have also been investigated [136]. There have been reports of unpleasant side effects, decreased bioavailability, and decreased medication delivery to the target location with certain pharmacotherapeutic drugs [137]. Adequate energy storage is necessary to maintain nutritional balance, and in mammals, this is achieved by producing and storing triglycerides in white adipocytes [137]. White adipocytes produce hormones in addition to storing energy; leptin and adiponectin control the energy balance [138]. According to reports, white adipose tissue (WAT) in the region predicts metabolic health in obese people [139]. The risk of developing metabolic syndrome is higher in obese people with expandable intra-abdominal WAT (visceral adiposity) than in people with subcutaneous WAT (subcutaneous adiposity) [140]. The fact that visceral and subcutaneous adipose tissues differ noticeably may be the reason for their potential roles in preserving energy balance. Due to its depot and proximity to the stomach, visceral obesity may have negative implications. The ease with which free fatty acids can enter the portal circulation from visceral adipose tissue can disrupt liver function [141]. The adipose depots of people with MetS have unusually high levels of pro-inflammatory macrophages, hypertrophic adipocytes, hypoxia, and fibrosis [140]. Oxidative stress, or the overproduction of reactive oxygen species, is a consequence of obesity. Obesity is further encouraged by oxidative stress, which damages cellular components and causes inflammation [136,142].

## 10. MetS and COVID-19 Association

Since its outbreak in December 2019, the COVID-19 pandemic has had a negative impact on our lives and prompted the creation of medicines and preventatives to fight the illness [143–145]. Despite our modest progress toward a COVID-free future, thousands of people continue to contract the disease yearly. The illness of COVID-19 may be complicated in many cases. While most patients improve after experiencing manageable symptoms, a sizeable percentage of individuals experience harsh consequences that necessitate hospitalization/ventilation and frequently result in multiple organ failure and death. The widespread notion is that a body's microbial burden causes disease. Therefore, the primary treatment efforts focus on eliminating the pathogen in affected people. However, it must be well understood that SARS-CoV-2, which causes COVID-19, infects our bodies and, like many other viruses, leverages the host's regular physiological systems, including the metabolic system, to maintain its growth and pathogenicity [146]. Alterations in the immunological homeostasis at the cellular or tissue level have long been known to be detrimental in several illness. For COVID-19, it is inarguable to suggest that intrinsic host status is inextricably tied to the individual immunological system reaction and capabilities to combat the Coronavirus, which influences the former pathophysiology and the progression of the illness. This scenario was observed in one of the first reports from 2020, when an investigator studied the serum of Coronavirus infected individuals and observed alterations in many vital metabolic processes, such as tryptophan and nitrogen metabolism, and amino acid metabolic pathways. Several observational research investigations that utilized statistics from the first wave of the COVID-19 pandemic have drawn attention to how the patient's pre-existing metabolic syndrome—a condition characterized by obesity,

hypertension, diabetes, and/or abnormal cholesterol levels—affects vulnerability to SARS-CoV-2 infections, its extremity, and the progression of the illness. Chronic diseases known as MetS aggravate the underlying inflammatory state and damage a number of the organs targeted by the COVID-19 virus. As a result, it makes sense why an undergoing inflammatory condition or damaged organs are pre-disposed to having serious complications after SARS-CoV-2 infection. The information from 287 individuals admitted to medical facility in New Orleans, LA, between 30 March and April 5, 2020, were analyzed, and it was found that people with MetS had a 5-fold higher risk of needing ventilation and a 3.40-fold higher risk of dying from COVID-19, despite adjusting for variations in race, age, gender, and characteristics of MetS [147].

## 10.1. Diabetes and COVID-19

According to recent studies, COVID-19 and the onset of diabetes are correlated bidirectionally. Even though it is unknown whether diabetic persons are more likely to become infected with the COVID-19 virus, investigators found that people with hyperglycemia are more prone to severe COVID-19 symptoms. According to a recent observation, when the pathogen enters the host body, it recruits and infects immune cells known as macrophages and monocytes. The explanation for this is still not completely understood. The virus then modifies the metabolism of infected monocytes and macrophages, causing them to become habituated to consuming glucose due to hyperglycemia. This metabolic reprogramming aids the virus's development and survival inside the host body [148,149]. Additionally, in diabetes settings, virus-infected macrophages and monocytes render T cells, the human body's vital immune cells that combat pathogens, useless. The high quantities of pro-inflammatory chemicals produced by the metabolically altered monocytes and macrophages cause patients to have an uncontrolled inflammatory response known as a cytokine storm. A storm of cytokines may cause quick and fatal damage to tissues and organs by overloading the human circulatory system with molecules that promote inflammation, and is frequently challenging to treat in hospitals. When blood glucose levels are elevated, the virus destroys lung epithelial cells, impairing lung function. Hence, COVID-19-induced metabolic flip and immune cell debilitation creates the way for serious illness manifestations and, in few situations, mortality [150]. T2DM raises the risk of getting diabetic kidney disease, which impairs kidney function and causes toxic substances to accumulate in the body. COVID-19 infection has also been related to kidney impairment. As a result, a person with a history of diabetes who already has compromised kidney function is more susceptible to developing severe renal impairment after a viral infection [151]. Additionally, physicians have noticed an increase in the number of COVID patients who have been diagnosed with either type 1 (the body cannot produce insulin) or type 2 (the body produces insufficient amounts of insulin) [152]. Postmortem reports from COVID-19 victims showed the virus might have spread to the pancreas. Additionally, the infected pancreas displayed areas of dead cells and significant immune cell infiltration (necrosis). This implies that the virus may harm the pancreas, including the beta cells, either directly or indirectly, raising the chance of developing diabetes or other dysfunctions, such as pancreatitis. According to a meta-analysis of 3700 individuals, 14.4% of patients hospitalized for severe SARS-CoV-2 symptoms acquired diabetes. The patients with "new diabetes" were more likely to need urgent treatment than those who already had diabetes before contracting the virus [153,154]. Even three years after the pandemic, researchers are still unsure whether COVID-19 speeds up the development of diabetes, causes it, or does both. Many of these people were diabetes-free at the time. After the virus had cleared, some patients with high blood sugar during the illness had no subsequent signs. However, some individuals received a diagnosis of a severe form of diabetes.

## 10.2. Obesity and COVID-19

Obesity has been linked to conditions such as diabetes, stroke, irregular heartbeat, and several types of cancer. It now has a connection to COVID-19 results. Former is

currently recognized as a cause for developing a significant COVID-19 illness, as per several researchers from the USA and other nations. A total of 84 of the 124 individuals treated in critical care with mechanical ventilation in French research, had high BMIs and were obese. In total, 90% of those with BMIs more than 35 required occasional obligatory ventilation. The immunological profile that indicates worse COVID-19 illness and BMI are also correlated. Low-grade inflammation is brought on by obesity. Immune cells in obese people have already been activated, and this activation is harmful. For instance, even if the individual is not infected by any bacterial or viral pathogens, the body of an obese person produces increased levels of inflammatory chemicals from immune cells, such as macrophages. T cells, another type of immune cell vital to battling infections or disease, are metabolically changed and functionally compromised in an obese individual. By harming the blood vasculatures through fat deposition and plaque formation, obesity raises the chances of thrombosis (clotting in specific areas of the circulatory system). It also increases the likelihood of respiratory impairment, chronic obstructive lung disease, and pulmonary fibrosis. Patients infected with SARS-CoV-2 are in danger of coagulation. Thus, virus-induced clots and pre-existing clots increase the chance of having a stroke or embolism. Obesity may raise physical stress on ventilation by limiting diaphragm excursion, influencing the treatment outcome [155,156].

### *10.3. Hypertension and COVID-19*

It is now evident that individuals with hypertension are more likely to have severe SARS-CoV-2 symptoms. According to initial observations from China and the United States, hypertension was the pre-existing disease in 30–50% of COVID patients hospitalized. According to an Italian survey, 76% of COVID-19 fatality patients had high BP. People of African American, Native American, or Hispanic ancestry are more likely to have hypertension, as well as more problems, hospitalizations, and fatalities associated with COVID-19 [157,158]. Hypertension causes several changes in our bodies over time, including damage to the blood vessels, resulting in chronic inflammation all across the body and reducing the immunological system's capacity to combat external invaders. Kidney injury is caused by damaged blood arteries, which reduce blood supply to the kidneys [157,158]. Given that the SARS-CoV-2 pathogen particularly targets the kidneys, it is not unusual to predict that individual with hypertensive renal disease is more prone to suffer from COVID-related kidney problems. As a result, it stands to reason that those who already have a disease that lowers their immunity or body's defenses will have a more difficult time battling the virus and will either be unable to remove it or experience harmful immune system hyperactivation.

### 11. Conclusions

MetS is a prevalent cause of global morbidity and mortality that has been linked to a variety of factors. Among the various pathophysiological mechanisms proposed for MetS, IR, central obesity, low-grade chronic inflammation, pro-inflammatory status, and oxidative stress are thought to be the most prevalent. To identify the absolute target of the therapy, further research into the underlying pathophysiology and factors linked with MetS is required. Emerging research on metabolomics and proteomics should be well standardized among the research labs to obtain more conclusive findings on different etiological factors related to MetS. Nevertheless, available investigations on the association of COVID-19 and MetS suggest additional works regarding the impact of the virus on patients already suffering from MetS for future preparedness to combat all sorts of pandemic threats in the future.

**Author Contributions:** Conceptualization, B.K.J., S.K.J., M.L.S. and K.R.P.; writing—B.K.J., M.I. and L.A.J.; writing—review and editing, Y.M. and M.L.S.; supervision, S.K.J. and K.R.P. All authors have read and agreed to the published version of the manuscript.

**Funding:** This research received no external funding.

**Institutional Review Board Statement:** Not applicable.

**Informed Consent Statement:** Not applicable.

**Conflicts of Interest:** The authors declare no conflict of interest.

## Abbreviations

AHA: The American Heart Association; NCEP: ATP III: The National Cholesterol Education Program Adult Treatment Panel III; NHLBI: The National Heart, Lung, and Blood Institute; BMI: body mass index; IDF: The International Diabetes Federation; EGIR: The Europe Group for the Study of Insulin Resistance; WHO: The World Health Organization; WC: west circumference; IR: insulin resistance; FBS: fasting blood sugar; TG: triglycerides; HDL-C: high-density lipoprotein–cholesterol; BP: blood pressure; Rx: on medication; T2DM: type 2 diabetes mellitus.

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
