# Peer review of "Progress in Understanding Metabolic Syndrome and Knowledge of Its Complex Pathophysiology"

_diabetology, doi:10.3390/diabetology4020015_

Round 1
Reviewer 1 Report
This article aims to review progress in metabolic syndrome knowledge, with a focus on its pathophysiology.
However, I have some concerns about this manuscript being considered for publication.
Major issue:
The content of the review is not original and does not give a new perspective on the topic.
Although the authors’ purpose was to review the pathophysiology of metabolic syndrome and the latest news about it, I found this paperlacking in that purpose.
I recommend that the authors expand the description of the pathophysiology of the disease, emphasizing its clinical relevance in the context of cardiovascular risk.
(See also Hsu CN, Hou CY, Hsu WH, Tain YL. Early-Life Origins of Metabolic Syndrome: Mechanisms and Preventive Aspects. Int J Mol Sci. 2021 Nov 2;22(21):11872. doi: 10.3390/ijms222111872. PMID: 34769303; PMCID: PMC8584419; Fahed G, Aoun L, Bou Zerdan M, Allam S, Bou Zerdan M, Bouferraa Y, Assi HI. Metabolic Syndrome: Updates on Pathophysiology and Management in 2021. Int J Mol Sci. 2022 Jan 12;23(2):786. doi: 10.3390/ijms23020786. PMID: 35054972; PMCID: PMC8775991.).
Minor issues:
The paper needs for an English language editing.
Authors should check the entire manuscript for abbreviations and capital letters.
Author Response
We have made the necessary changes as suggested by the reviewer

Reviewer 2 Report
The authors should be carefully with the abbreviation MS for metabolic syndrome, because the MS is mostly used for multiple sclerosis as another autoimmune disease.
The tables are not easy to understand. The authors should underlay similarities or differences in grey values.
In table 3 similarities in the threshold in abdominal obesity of the countries could be collected together.
In Fig. 1 the scheme of visceral obesity should include all fat depots in the abdominal region e.g. between the vessels for the blood supply of the different organs.
Fig.2 is the glucose uptake in the periphery of the organism e.g. muscle?
The legend of Fig. 3 a word like tissue is missing.
Author Response
We have made the necessary changes as suggested by the reviewer.
